# Multi-Agent Reinforcement Learning for Traffic Signal Control: A Cooperative Approach

Máté Kolat [ID], Bálint Kővári [ID], Tamás Bécsi *[ID] and Szilárd Aradi *[ID]

Department of Control for Transportation and Vehicle Systems, Budapest University of Technology and Economics, H-1111 Budapest, Hungary
* Correspondence: becsi.tamas@kjk.bme.hu (T.B.); aradi.szilard@kjk.bme.hu (S.A.)

**Abstract:** The rapid growth of urbanization and the constant demand for mobility have put a great strain on transportation systems in cities. One of the major challenges in these areas is traffic congestion, particularly at signalized intersections. This problem not only leads to longer travel times for commuters, but also results in a significant increase in local and global emissions. The fixed cycle of traffic lights at these intersections is one of the primary reasons for this issue. To address these challenges, applying reinforcement learning to coordinating traffic light controllers has become a highly researched topic in the field of transportation engineering. This paper focuses on the traffic signal control problem, proposing a solution using a multi-agent deep Q-learning algorithm. This study introduces a novel rewarding concept in the multi-agent environment, as the reward schemes have yet to evolve in the following years with the advancement of techniques. The goal of this study is to manage traffic networks in a more efficient manner, taking into account both sustainability and classic measures. The results of this study indicate that the proposed approach can bring about significant improvements in transportation systems. For instance, the proposed approach can reduce fuel consumption by 11% and average travel time by 13%. The results of this study demonstrate the potential of reinforcement learning in improving the coordination of traffic light controllers and reducing the negative impacts of traffic congestion in urban areas. The implementation of this proposed solution could contribute to a more sustainable and efficient transportation system in the future.

**Keywords:** deep learning; reinforcement learning; air pollution; road traffic control; multi-agent systems

## 1. Introduction

With the rapid development of industrial technologies and the economy combined with the improvement of the general quality of life, the level of mobility greatly increased in recent decades. According to research [1] conducted in Delhi, the number of certain types of vehicles tripled by 2020 compared to 1990, bringing great pressure into the urban road network. Traffic congestion raises several major problems, such as unnecessary resource loss, when people waste a significant amount of time while waiting in traffic jams, and also causing significant environmental damage locally and globally by releasing a massive amount of pollutants into the atmosphere. A research in Finland [2] stated that the average temperature between 1847 and 2013 raised by 2.4 °C, but its acceleration is even more shocking, as it tripled in the last 40 years. Despite the stricter emission standards and laws, the amount of harmful substances emitted continues to grow every year. One of the solutions is to reduce the road network load; consequently, the environmental load in the urban area is designing accessible public transportation, which avoids the need to use a car in the city area. In [3], the authors investigated the impact of subway accessibility based on four different strategies. Electric shared mobility can also provide a solution to reduce the environmental load in urban areas. However, charging electric vehicles is only available for

some people. In [4], the authors investigated the influencing factors on drivers' charging station selection, providing important implications for planning charging infrastructure and managing demand for city-wide electric vehicles. Another direct way to tackle the traffic congestion problem is to control intersections. Due to the lack of real-time data, the most common traffic control method globally is the fixed-cycle program, where the phases of the traffic light switch between pre-defined intervals. Acquiring a new range of road data became available through the sustainable development of wireless communication in recent years, allowing for new traffic light control solutions, such as adaptive traffic signal control (ATSC). ATSC makes it possible to change traffic lights according to the current state of the congestion based on different metrics. This paper presents a multi-agent reinforcement learning-based ATSC solution, utilizing deep Q-learning and considering a six-intersection road network. The proposed method reduces the congestion of the randomly filled road network while being evaluated by classic waiting time, travel time and sustainability measures.

### 1.1. Related Work

TSC is a widely researched topic, resulting in various studies, which can be distinguished considering the complexity of the network—single or multiple intersections—or the type of the approach, e.g., classic method or machine learning. The Webster technique [5] is a classic approach applied to a single intersection, assuming that the traffic flow is uniform during a specific period and calculating an optimal phase split and cycle length, aiming to minimize the travel time. The Greenwave method [6] aims to optimize by reducing the number of stops in one direction by sharing the intersections with the same cycle length. Thus, Greenwave optimizes only for unidirectional roads. The Max-pressure control [7] prevents an intersection from over-saturation by optimizing the pressure, which is the difference between the queue length of the incoming and outbound lane. In contrast, SCATS [8] selects from pre-defined signal plans based on pre-defined performance measurements. However, in many of these cases, the research assumes the traffic flow is uniform and constant, which does not describe the real world correctly. In contrast, machine learning methods are able to learn and tune their strategy by relying only on environmental observations and feedback, without any assumptions, which is an enormous benefit compared to the classic methods. In RL, the feedback from the environment is called reward. In the case of TSC, the classic rewarding methods use measurable components or its weighted linear combination. In [9], the author utilizes queue length to design the rewarding system. In [10], the author introduces a reward, which uses waiting time, while [11] presents a method where the reward includes the delay of the vehicles. Regarding state representation, there are image-based approaches in TSC as well. In [12], a pixel-value-based method is introduced. This technique utilizes a convolutional neural network (CNN), where the lanes are divided into cells, while each cell is interpreted as a pixel. In [13], a model-free graph method is presented, where a graph convolutional neural network (GCNN) extracts geometry road network features and adaptively learns the policy. In [14], the authors propose a two-game theory-aided RL TSC algorithm leveraging Nash equilibrium and reinforcement learning. In [15], the authors propose the PDA–TSC method, introducing mixed pressure, which enables RL agents to simultaneously analyze the impacts of stationary and moving vehicles on intersections. Finally, in [16] the authors present a spatio-temporal multi-agent RL (STMARL) framework for multi-intersection traffic light control. This paper introduces a MARL system with a novel rewarding strategy and compares the results to the fixed-cycle method and other reinforcement learning techniques.

### 1.2. Contribution

Many papers address the TSC problem in a MARL environment; however, most utilize a classic rewarding method derived from the objective functions of the classic optimization or control-based methods, where the designed system is compared to a fixed-cycle traffic control system. This paper introduces a novel rewarding technique, which is the follow-

up of a previous research [17] in a MARL environment. This study utilizes a reward strategy, where the standard deviation of the vehicle's halting number of the incoming lanes in each intersection is considered. This paper shows the applicability of this novel rewarding method in a multi-agent environment, as it outperforms the policy gradient (PG)-based trained agent from the previous research and the baseline controller in classic and sustainability measures.

Moreover, this solution outperforms the actuated control as well, which extends the traffic phase when a continuous traffic flow is detected. After detecting a sufficient time interval between successive vehicles, a switch to the next phase occurs. This control allows for a better distribution of green time across phases and the ability to influence cycle times according to dynamic traffic conditions. It is significant to compare this solution with actuated control, which is commonly used in Germany because it provides a benchmark for its performance. Actuated control is widely accepted and used, so comparing the proposed solution with it highlights its performance and demonstrates the proposed solution's potential. By outperforming a widely used control scheme, it showcases the potential of the proposed solution to bring about significant improvements in traffic flow control and congestion reduction. This comparison strengthens the argument for implementing the proposed solution in real-world transportation systems and highlights its potential for widespread adoption and impact. The results present that further emission and travel time reduction are reachable with this rewarding strategy.

## 2. Methodology

This study presents an adaptive TSC method, applying multi-agent reinforcement learning (MARL) with deep Q-learning. Section 2.1 introduces the literature background of reinforcement learning (RL). Section 2.2 presents the basis of MARL, while Section 2.3 shows the motivation for deep-q learning.

### 2.1. Reinforcement Learning

Although reinforcement learning has reached success in the past [18–21], previous approaches faced several limitations such as dimensional, memory, computational or sample complexity issues [22]. The rise of the deep neural network provided new tools to solve these problems. Deep learning significantly improved several areas in machine learning, such as object detection, language translation or speech recognition. The application of deep learning in RL defined a new field called deep reinforcement learning (DRL). In 2015, a DRL algorithm was developed to play Atari 2600 games on a superhuman level based on gameplay images [23]. This was the first research proving that a neural network could be trained on raw, high-dimensional observations, i.e., on unstructured data based solely on a reward signal. They were followed by developing an RL algorithm that defeated the human world champion in Alpha GO [24].Since then, RL has been applied in many applications, from robotics [25] through video games [23] to navigation systems [26]. A core topic in machine learning (ML) is sequential decision-making. The main task of this topic is to decide from experience the sequence of actions that allows for performing in an uncertain environment to reach some objectives. The main principle of RL is shown in Figure 1.

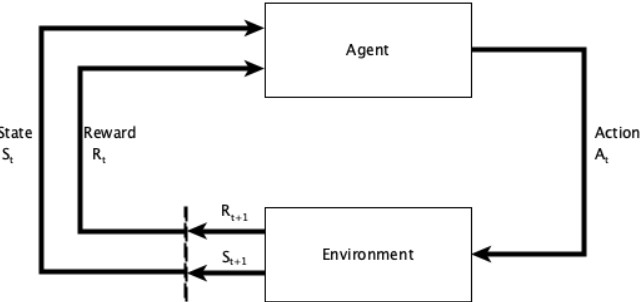

**Figure 1.** The reinforcement learning basics.

A decision-maker, called an agent, observes the environment and takes actions based on its actual state. Each action is evaluated by its quality. The agent is penalized or rewarded based on whether its action is unwanted or desirable. RL is modeled on the Markov decision process (MDP), defined in terms of <S, A, R, P>:

- S: Set of observable states;
- A: Set of possible agent actions;
- R: Set of gathered rewards based on the quality of the action;
- P: The policy is to decide which action is selected at a given state.

An adequately designed rewarding strategy in RL highly affects the performance of tuning the agent's neural network (NN) to reach the desired behavior. This type of learning aims to adjust a network, generalizing its responses to perform correctly in unknown environments that are not included in the training set.

### 2.2. Multi-Agent Reinforcement Learning

There are complex scenarios where an individual agent is insufficient to solve a specific problem. In this case, a so-called multi-agent reinforcement learning (MARL) system can be introduced, where multiple interacting agents share a common environment. However, the MARL system's efficiency is highly dependent on the coordination of the agents. The MARL technique, based on the coordination strategy, can be categorized in different manners. In [27], the author distinguishes two classes, namely independent learners (ILs) and joint action learners (JALs). ILs apply Q-learning in a classic sense without considering the other agents. In contrast, the JALs' approach is to learn the value of their own actions considering the other agents via the integration of RL with equilibrium learning methods.

In the case of TSC, these categories can be extended. In [28], the research introduces a centralized approach where a single global agent is used for multi-intersection scenarios. It takes the state of all intersections as input and learns to select the actions for all intersections simultaneously. However, this method faces dimensionality problems as the state space grows exponentially with the number of intersections. In [29], the author utilizes coordination graphs, considering direct coordination mechanisms between the learning agents, while factorizing the global Q-function as a linear combination of local sub-problems as follows:

$$\hat{Q}(s_1,...,s_N;a) = \sum_{i,j} Q_{i,j}(s_i, s_j, a_i, a_j) \tag{1}$$

Two subtypes of ILs can be differentiated in RL. One is an IL without communication. In this case, there is no explicit coordination between the agents. The observation of agent *i* depends on only the state of intersection *i*. This technique is sufficient in simpler networks or arterial roads, forming mini-green waves. However, in the case of more complex environments, the impacts of the traffic from the neighboring intersections are being brought into the environment, preventing the learning process from converging if there is no direct communication or coordination among the agents [30]. Communication between the agents can eliminate this problem. Coordination and communication are highly important, especially in TSC, because the intersections are close to each other in real-world scenarios and the traffic is very dynamic [31].

### 2.3. Deep Q-Learning

Q-learning is an off-policy action-value-based reinforcement learning algorithm. It aims to find the best course of action based on the environment's state. A deep Q-network (DQN) utilizes a Q-learning framework with a neural network (NN). Deep Q-learning utilizes two neural networks during the learning process, namely the main network and the target network. These networks are structured the same but with different weights. The main network ($Q(s,a;\theta)$) estimates the recent Q-value and contains all updates during learning, while the target network ($Q(s,a;\theta^-)$) retrieves the old parameters to estimate the

next Q-value. In every $n_{th}$ step, the weights of the main network are copied to the target network. Without a target network, the training would become unstable; thus, using two networks helps to reach a more stable and effective training process. The first step of deep Q-learning is to initialize the weights of these two networks. In the second step, the actions are selected based on the epsilon-greedy exploration strategy. This technique helps the algorithm to avoid becoming stuck in the local maximum by choosing random actions with the probability $\epsilon$ and exploits the action with the highest value with the probability of $1 - \epsilon$. Afterward, the weights of the main network are updated according to the DQN loss using a random set of the experience replay.

Experience replay is a replay technique storing the agent's experience $< S, A, R, S' >$ in every step over many episodes. It allows for the sampling of a randomly selected mini-batch from memory, which helps avoid the auto-correlation problem leading to unstable training. The calculation of the DQN loss is shown in Figure 2 and (2), where it is calculated as the difference between the Q-value of the main network and the target network.

$$L_i(\theta_i) = \left( \underbrace{r + \gamma \max_{a'} Q(s', a'; \theta_i^-)}_{target} - \underbrace{Q(s, a; \theta_i)}_{predicted} \right)^2 \tag{2}$$

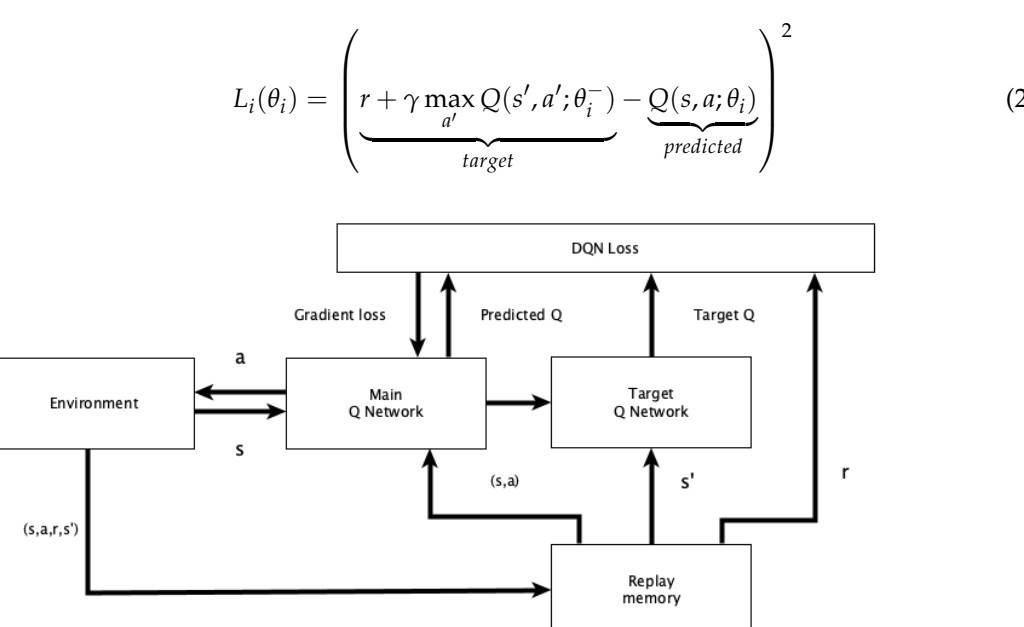

**Figure 2.** The calculation of DQN loss [32].

### 2.4. Policy Gradient Algorithm

While Q-learning is a typical action-value-based off-policy method that aims to optimize the Q value, the policy gradient method seeks to optimize the policy directly by training the network weights using gradient descent. Hence, the output of the network is a probability distribution over the possible actions $\pi$, specifying the agent behavior. The probabilities of actions do not carry information regarding their long-term consequences but only indicate what is necessary for the given scenario. Compared to value-based RL methods, PG methods have the advantage of guaranteed convergence at least to a local optimum [33]. The update rule for the weights is as follows:

$$\theta \leftarrow \theta + \alpha \nabla \log \pi_\theta(s_t, a_t) \sum_{t=1}^{\tau} \gamma^t r_t \tag{3}$$

where $\pi_\theta(s_t, a_t)$ is the probability of action $a$ in, state $s$. $\alpha$ is the learning rate, affecting the step size of the gradients during the training.

The update steps are:

1. The neural network's weights $\theta$ are initialized at the start of the training. Subsequently, the training starts from an initial state $s_0$;

2. The interactions between the agent and the environment $< S, A, R, S', D >$ are saved into the episode history. The interactions continue until a terminal condition is reached;

3. After the termination occurs, the cumulated and discounted reward is calculated based on the interactions stored in the episode history;

4. Finally, the gradient buffer is extended by the calculated gradient, which is used for updating the neural network's weights based on the chosen update frequency.

### 2.5. Baseline Controller

The baseline controller uses a fixed phase and fixed cycle traffic light control. The available phases are East–West Green and North–South Green, while the phase length is set to 30 s, and the changes are separated with 3 s of the yellow phase.

### 2.6. Actuated Controller

The applied actuated controller is a built-in algorithm in the SUMO simulation software that continuously monitors incoming traffic, considers the cumulative time delay of each vehicle and modifies traffic lights accordingly.

## 3. Environment

This research used SUMO (Simulation of Urban MObility) microscopic multi-modal road simulation software for environment design. SUMO contains many traffic modeling utilities, such as different demand generation and routing, allowing for a diverse training dataset. Moreover, the simulation software provides various measures; therefore, a diverse set of reward and state-representation designs can be compared, allowing for the selection of the most suitable for the chosen problem. Furthermore, SUMO includes a remote control interface called TraCI (Traffic Control Interface), allowing users to retrieve data from simulation objects and control their behavior online, allowing for the agent and the implemented environment to communicate. The designed road network shown in Figure 3 contains six intersections. Each road segment is a tidal flow road with a length of 500 m, with a width of 3.5 m. Traffic lights control the traffic at each intersection. The traffic demand and routing are randomized in each training episode, providing a diverse dataset. The period time for spawning vehicles is designed by a set of probabilities. The built-in function in SUMO is utilized for evaluation purposes, which can return with the impact of the traffic, such as emitted pollutants.

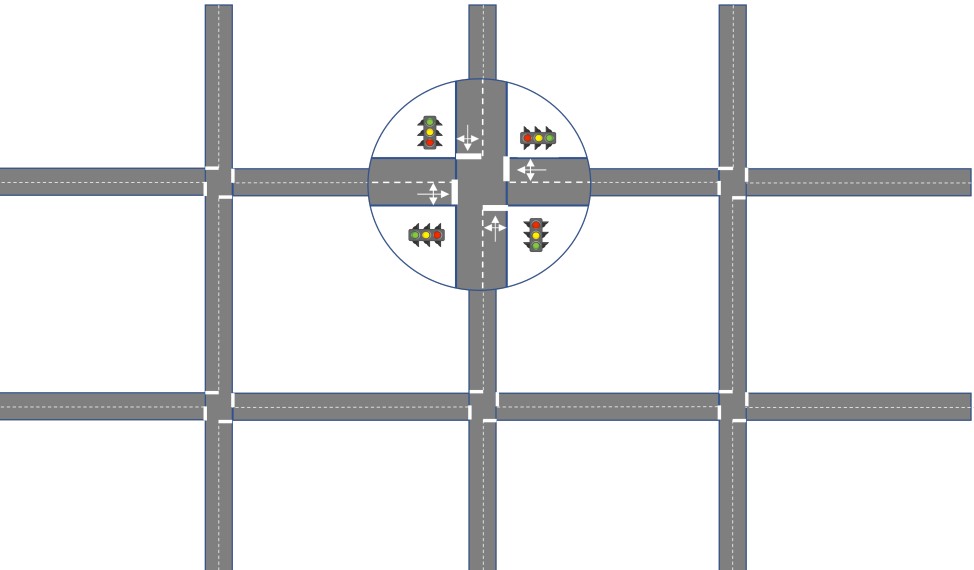

**Figure 3.** The designed road network.

### 3.1. State Representation

Inadequately chosen state-space representation data can easily block the network from being trained optimally. For the TSC problem, two different state-space representations can be distinguished. In the first case, the state representation is a feature-based vector, while the other is distinguished as an image-like representation. In this study, the feature-based vector method is utilized, as all the necessary information is available from SUMO. This paper presents a state-space containing the halting number of all incoming lanes in a percentage form and the number of the current intersection in binary. The advantage of utilizing a halting number over occupancy is that a halting number contains only the number of vehicles with speeds less than 0.1 $\left[\frac{m}{s}\right]$. Thus, diverse types of noises can be filtered out, such as the noise of newly spawned vehicles. Since each intersection's agent sees the data of all lanes, it is necessary to determine which agent of the intersection makes the current decision; therefore, the state-space representation must contain the number of the current intersection.

### 3.2. Action Space

In this study, two discrete and valid actions are differentiated. One is the so-called North–South Green, when vehicles approaching the intersection from the north or south lane get the green light. The other action is the East–West Green, which, as its name suggests, gives the green light to vehicles arriving from the east and the west. From each lane, the allowed maneuvers are the right turn, left turn and the straight-ahead, which the rules of the traffic code can carry out. The length of the red and green phases is fixed to 30 *s*; however, the phase change frequency is not predetermined. Each phase is separated with yellow signals, which last for 3 s. As the traffic is generated randomly in each episode, the agent's goal is to find the best possible phase based on the actual state.

### 3.3. Reward

A well-designed reward function is crucial in reinforcement learning, as this metric is the only feedback the agents obtain based on its action from the environment. One can define the reward function arbitrarily; however, in the case of a TSC problem, a few literature solutions are commonly used, such as queue length, average speed, waiting time, delay or time lost [34,35]. The concept of the utilized rewarding system is that not only the basic queue length but its standard deviation is considered. Thus, the standard deviation of the four road segments is calculated at each intersection. At the same time, the agent's goal is to minimize the distribution's standard deviation ($\sigma$) of the vehicle's halting number. This solution is also better than the simple queue length solution, as it also considers the vehicle distribution of the intersection, giving an even more accurate picture of the current traffic situation.The core of this idea is in the new optimization problem that the abstractions formulate for the agent. The state representation is the density of the links scaled to a [0,1] interval. This is considered a mini-distribution. The mean and the standard deviation of the distribution can be controlled through actions, since the actions can simply reduce or increase the density of the links. Consequently, the goal of the proposed reward function is to minimize the standard deviation of the distribution formulated from the densities of the links in the network. This objective naturally supports generalization since it teaches the agent that the reward can only be increased if the density differences are resolved in the network. A minimal standard deviation of the densities means that despite the asymmetric traffic loads, the queues in the intersections are the same. This paper utilizes a cooperative MARL technique with an immediate rewarding method. Each agent obtains its reward based on the distribution of the particular intersection after each action. The highest achievable reward is 1, given to the agent when the standard deviation is zero, meaning that every incoming lane is loaded equally. Consequently, the agents are keen on equalizing the drivers' individual waiting times, resulting in greater satisfaction among road users. The rewards determined on the basis of standard deviation are presented in Table 1.

**Table 1.** Reward parameters.

| $\sigma_{min}$ | $\sigma_{max}$ | Minimum Reward | Maximum Reward |
|---|---|---|---|
| 0 | 0.1 | 0 | 1 |
| 0.1 | 0.5 | −1 | 0 |

If $\sigma < 0.1$, then the parameters are the following: $\sigma_{min} = 0$, $\sigma_{max} = 0.1$, $R_{min} = 0$ and $R_{max} = 1$. In contrast, when $\sigma \geq 0.1$, $\sigma_{min} = 0.1$, $\sigma_{max} = 0.5$, $R_{min} = -1$ and $R_{max} = 0$.

### 3.4. Training

During training, the episodes start with a warm-up phase, which lasts until the road network's occupancy reaches 10%. Each episode uses a different random seed to keep diversity. During warm-up, no action is taken by the agents, and no reward is given. After the warm-up phase, the state of each intersection is gathered and provided to the shared neural network in every step. Based on this information, an action is taken in each intersection, and the given reward judges its quality. Each training episode ends when the last vehicle leaves the road network or the 1800th episode step is exceeded.

## 4. Results

The results are evaluated based on two different measures. Firstly, the classic metrics are introduced: the waiting and travel time. Then, the sustainability measures are presented, showing the reduction in the pollutants emitted by the traffic utilizing the given technique. The metrics are compared between four different controls—the presented MARL control, a control where a single agent was trained on a single intersection, utilizing the same novel rewarding system, the actuated controller and the baseline controller. The proposed solution was tested using a total of 10,000 testing episodes. Each episode was run with a different randomly generated traffic flow, simulating real-world traffic conditions. This testing procedure was designed to evaluate the effectiveness of the control strategy developed during the training phase. The results show that the proposed solution could significantly reduce, e.g., fuel consumption and travel time compared to traditional control methods. Using random traffic flow in the testing phase ensured the solution was robust and could handle diverse traffic conditions effectively.

### 4.1. Training Process

Since the main bottlenecks of all training are the hyperparameters, a random grid search was conducted to find the best for both algorithms. The main parameters for both algorithms are listed in Tables 2 and 3.

**Table 2.** The training parameters of the DQN algorithm.

| Parameter | Value |
|---|---|
| Learning rate ($\alpha$) | 0.0003 |
| Discount factor ($\gamma$) | 0.95 |
| Batch size | 512 |
| Experience memory size | 15,000 |
| Num. of hidden layers | 2 |
| Num. of neurons | 256,256 |
| Hidden layers activation function | RELU |
| Method | DQN |
| Freq. of weight sharing (in ep.) | 10 |
| Layers | Dense |
| Optimizer | Adam |
| Kernel initializer | Xavier normal |

**Table 3.** The training parameters of the PG algorithm.

| Parameter | Value |
|---|---|
| Learning rate ($\alpha$) | 0.00005 |
| Discount factor ($\gamma$) | 0.97 |
| Num. of ep. after params are upd. ($\xi$) | 20 |
| Num. of hidden layers | 4 |
| Num. of neurons | 128,256,256,128 |
| Hidden layers activation function | RELU |
| Layers | Dense |
| Optimizer | Adam |
| Kernel initializer | Xavier normal |

*4.2. Classic Performance Measures*

The average travel time is calculated based on the total time spent by the vehicles on the road network and the number of participants, calculated as follows:

$$\bar{t}_t = \frac{1}{N} \sum_{i=0}^{N} (t_i end - t_i start) \tag{4}$$

where $N$ is the number of vehicles entered the road network, $t_i end$ is the time when $i_{th}$ vehicle leaves the network and $t_i start$ is the time when $i_{th}$ vehicle enters.

The other evaluated classic metric is the average waiting time, calculated by the sum of the accumulated waiting of all participants divided by its number, described as:

$$\bar{t}_w = \frac{1}{N} \sum_{i=0}^{N} t_i w \tag{5}$$

The results, which are presented in Table 4, show that the utilization of the introduced novel rewarding method in MARL further improves the system's performance. The PG agent, trained in a single intersection environment, shortened the travel time by 8%, corresponding to the baseline SUMO controller. Compared to the PG agent, an additional 9% of improvement has been achieved by utilizing multiple agents. In the case of average waiting time, the multi-agents system outperforms the PG agent by 15%, the actuated control by 19% and the baseline controller by 24%.

**Table 4.** Classic measures.

| Agents | Avg. Waiting Time [s] | Avg. Travel Time [s] |
|---|---|---|
| DQN–MARL | 236.28 | 507.74 |
| PG–SA | 275.72 | 556.26 |
| Actuated | 289.68 | 578.87 |
| Fixed cycle | 307.68 | 602.33 |

Figure 4 presents the controllers' performances considering average waiting time and average travel time during the 10,000 evaluation episodes.

The DQN–MARL agent, which utilizes our rewarding method, is shown in purple, the PG–SA controller is presented in black, the actuated control is described by the red color and the SUMO fix-cycled baseline controller's result is colored in yellow. The left figure illustrates the participants' average waiting time (s), which appeared on the road network. The right figure shows the average travel time (s) in the same structure. It can be seen that the MARL-based method outperforms the other solutions in each traffic scenario. From the human point of view, the vital aspect of the classic measure is that the reduction in waiting and travel time can significantly create a more calm traffic environment, decreasing the number of irritated drivers.

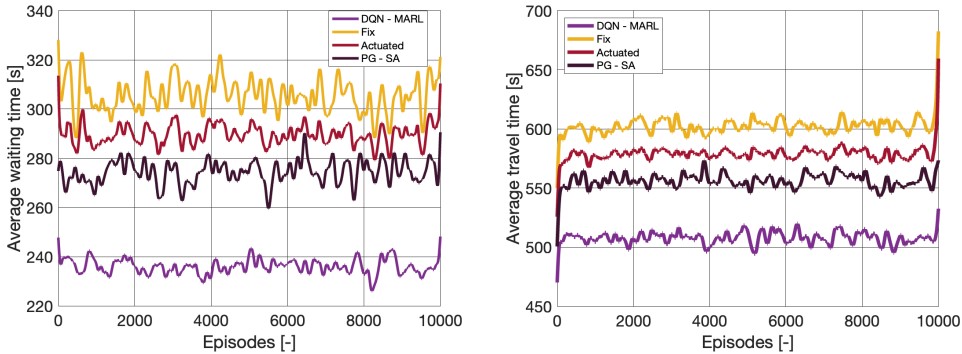

**Figure 4.** Classic measure performance of different controllers during the evaluation episodes.

*4.3. Sustainability Performance Measures*

Table 5 presents the performance of different solutions for the TSC problem, regarding the sustainability measures. The emissions are calculated by the SUMO software in each time step utilizing the mathematical model from *The Handbook Emission Factors for Road Transport* [36].

**Table 5.** Sustainability measures.

| Agents | $CO$ [kg] | $CO_2$ [kg] | $NO_x$ [g] | *Fuel*[l] | $HC$[g] | $PM_x$[g] |
|---|---|---|---|---|---|---|
| DQN–MARL | 59.77 | 1143.75 | 553.15 | 503.21 | 302.25 | 28.59 |
| PG–SA | 67.32 | 1219.63 | 596.47 | 536.63 | 338.99 | 31.35 |
| Actuated | 70.85 | 1286.81 | 612.39 | 566.19 | 352.68 | 32.36 |
| Fixed cycle | 75.37 | 1356.97 | 651.52 | 597.04 | 378.62 | 34.33 |

This shows that the proposed method outperforms the others in these aspects. Among the greenhouse gases, the new technique dramatically reduces the $CO_2$ emissions—one of the leading causes of global warming—compared to the other solutions. A decrease of 7% can be observed compared to the PG agent, 11% to the actuated controller and 15% to the baseline controller. The other critical pollutant is $NO_x$, which is highly emitted by diesel vehicles and is the reason for acid rain. Here, a minimum of 12% emission reduction is reached. Considering $CO$ emissions, around 11% of reduction is reached compared to the PG agent, 16% to the actuated controller, while 21% to the SUMO fix-cycle controller. At least an 8% $PM_x$ drop is observable. At the same time, in the case of $HC$, the performance of the control utilizing the novel rewarding system outperforms the PG agent by 11%, the actuated controller by 14% and the baseline controller by 21%. Lastly, a significant—minimum of 7% of fuel savings is achieved. Hence, the sustainability measures prove that a significant improvement in emissions is achieved with the proposed method.

Figure 5 presents the controllers' performances considering $CO_2$ emission and fuel consumption during the evaluation episodes. The different controls are presented in the same manner, as it was mentioned above. The left figure presents the cumulative $CO_2$ emitted by the participants, which appeared on the road network. The right figure shows the fuel emission in the same form. The table validates the utilization of the standard deviation-based reward with multiple agents, showing that the introduced controller significantly outperforms the PG–SA, the actuated controller and the baseline controller in sustainability metrics as well.

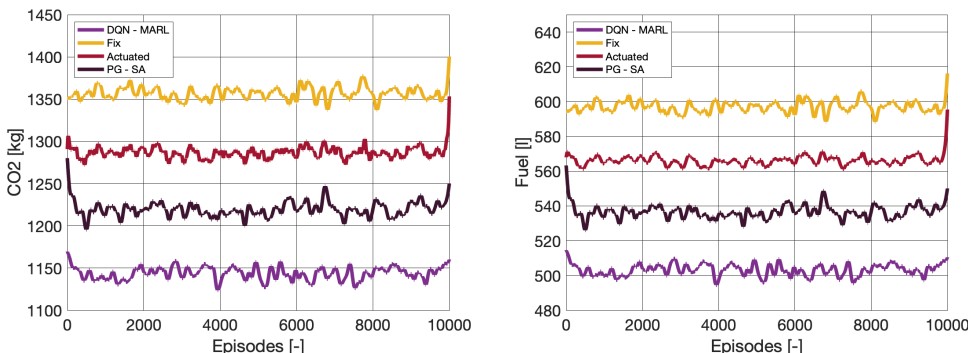

**Figure 5.** Sustainability measure performance of different controllers during the evaluation episodes.

## 5. Conclusions

This paper presents a cooperative MARL-based solution for the TSC problem, utilizing multiple trained agents in a six interconnected intersection environment for classic and sustainability metric reduction purposes. The solution's performance is compared to the SUMO's baseline controller and another trained agent, utilizing the same novel rewarding method. The results suggest that the proposed method outperforms the other controllers in every indicator; hence, this study verifies that adaptive control techniques can greatly reduce emissions of pollutants in urban areas and significantly decrease the time spent in traffic. Managing traffic jams is one of the biggest challenges in urban environments today. The presented method offers a quasi-infrastructure-independent solution, as the amount of traffic can be measured with the help of loop detectors which are an integral part of the road network nowadays. The sustainability measures' results greatly affect the global and local environment, as a higher average life expectancy can be reached due to the fewer inhaled pollutants in the urban area. However, the technique introduced in this study requires infrastructure development; therefore, a higher initial investment cost. For collecting the vital data for the state-space representation, loop detectors or traffic cameras are mandatory for calculating the traffic density. Moreover, for communication purposes, implementing V2X is required.

The behavior analysis of the suggested solution indicates that the performance can be further enhanced by implementing communication between the intersections, like utilizing historical data, such as the action of the other agents or the previous road network state. Accordingly, the future endeavors of this research will be concentrated on investigating such solutions to achieve even better performance in the case of even more complex transportation networks.

**Author Contributions:** Conceptualization, M.K., T.B. and Szilárd Aradi; Methodology, T.B.; Software, M.K., B.K. and S.A.; Validation, B.K.; Writing—original draft, M.K. and B.K.; Writing—review & editing, T.B. and S.A.; Visualization, B.K.; Funding acquisition, S.A. All authors have read and agreed to the published version of the manuscript.

**Funding:** This research was supported by the European Union within the framework of the National Laboratory for Autonomous Systems. (RRF-2.3.1-21-2022-00002). The research reported in this paper is part of project no. BME-NVA-02, implemented with the support provided by the Ministry of Innovation and Technology of Hungary from the National Research, Development and Innovation Fund, financed under the TKP2021 funding scheme. This paper was also supported by the János Bolyai Research Scholarship of the Hungarian Academy of Sciences.

**Institutional Review Board Statement:** Not applicable.

**Informed Consent Statement:** Not applicable.

**Data Availability Statement:** Not applicable.

**Conflicts of Interest:** The authors declare no conflict of interest.

## Abbreviations

The following abbreviations are used in this manuscript:

| | |
|---|---|
| ATSC | Adaptive traffic signal control |
| RL | Reinforcement learning |
| SCATS | Sydney coordinated adaptive traffic system |
| CNN | Convolutional neural network |
| GCNN | Graph convolutional neural network |
| MARL | Multi-agent reinforcement learning |
| STMARL | Spatio-temporal multi-agent reinforcement learning |
| PG | Policy gradient |
| DQN | Deep Q-network |
| DRL | Deep reinforcement learning |
| ML | Machine learning |
| MDP | Markov decision process |
| ILs | Independent learners |
| JALs | Joint action learners |
| NN | Neural network |
| SUMO | Simulation of urban mobility |
| TraCI | Traffic control interface |

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
