# Peer review of "Multi-Agent Reinforcement Learning for Traffic Signal Control: A Cooperative Approach"

_sustainability, doi:10.3390/su15043479_

Round 1

Reviewer 1 Report

Dear authors.

Thank you for your article. It is very interesting, and the results presented in it are promising (perhaps it will be possible to improve traffic thanks to them).

The layout of the entire document is correct. Your literature studies and research are clear. Therefore, I have no major comments to the manuscript.

My remarks:

- Abstract - consider adding a short summary of the results

- (line 12) - "mobilization" or "mobility"???

- Conclusion - consider adding a brief note about potential barriers of the application of your results in practice

Author Response

Thank You for your review. Please find our answers in the attached file.

Reviewer 2 Report

ATSC is a vast and systemic research field and has accumulated many fruitful outcomes since the last century. The reviewer considers that the current version of this paper needs to be revised significantly, with the following two major concerns.

The first is about innovation. The use of MARL to solve ATSC problems started five years ago and has gradually formed a trend, which is one of the main applications of RL in the transportation. It can be said that even MARL-based ASTC methods can now be organized as a review paper. The innovation stated in this paper is a reward function that considers intersection pressure deviation, which needs to be more. First, reward design in RL is a very engineering problem, which in most cases is formulated by ongoing trials and errors for a specific scenario, so how to ensure the generalizability of the pressure deviation reward to other ATSC scenarios? Second, the reviewer also doubts this idea of designing the reward function considering the deviation; why balancing the pressure can effectively reduce the overall traffic delay? What is the principle behind this? The reviewer thinks that the idea of balancing the pressure works solely on the random traffic flow scenario. The expected traffic flow is the same for each direction, so balancing the traffic pressure becomes the ideal solution for this specific scenario.

The second is about validity. As mentioned before, ATSC is a fruitful research area, and there are numerous adaptive control methods, including RL methods, and the authors should consider comparing state-of-art methods to illustrate the validity of the research. This paper only compared fixed signal timing and PG trained on a single agent, and such a comparison needs to be revised. Moreover, this paper uses the IQL algorithm, which is not strictly categorized under the MARL field. Finally, the size of the road network containing six intersections is relatively small and is far from the actual urban ATSC. Coordinating the signal control of a large-scale road network is one of the well-known hardships in this field. However, as aforementioned, there are many outcomes accumulated in this area, and if such critical unresolved problems are not focused on, the significance of this paper may be greatly diminished.

There are some other defects, such as redundancy in the introduction of basic knowledge, outdated literature, and lack of necessary discussion of agent training process. However, these are negligible compared to the above concerns because innovation and validity are fundamental to supporting this paper.

The authors can also expand their literature review effort to include the following publications: "Modeling the preference of electric shared mobility drivers in choosing charging stations" and "Subway Station Accessibility and Its Impacts on the Spatial and Temporal Variations of Its Outbound Ridership"

Author Response

Thank You For your review. Please find our answers in the attached document.

Reviewer 3 Report

This manuscript shows good research paper. However, some part need further improvement such Abstract, Findings and Discussion sections.

Author Response

Thank you for your review. Please find our answers in the attached document.

Round 2

Reviewer 2 Report

The authors have addressed all my concerns.